# Effects of structurally distinct human HDAC6 and HDAC6/HDAC8 inhibitors against *S. mansoni* larval and adult worm stages

**Roberto Gimmelli[1,2], Giuliana Papoff[1], Fulvio Saccoccia[1], Cristiana Lalli[1], Sandra Gemma[3], Giuseppe Campiani[3], Giovina Ruberti**[1]*

**1** Institute of Biochemistry and Cell Biology, National Research Council (IBBC-CNR), Adriano Buzzati-Traverso Campus, Monterotondo, Rome, Italy, **2** Dipartimento di Scienze Biochimiche "A. Rossi Fanelli", Sapienza Università di Roma, Roma, Italy, **3** Department of Biotechnology, Chemistry and Pharmacy, University of Siena, Siena, Italy

* giovina.ruberti@cnr.it

**Data Availability Statement:** All relevant data are within the manuscript and the supporting information.

## Abstract

Schistosomiasis is a major neglected parasitic disease that affects more than 240 million people worldwide caused by Platyhelminthes of the genus *Schistosoma*. The treatment of schistosomiasis relies on the long-term application of a single safe drug, praziquantel (PZQ). Unfortunately, PZQ is very effective on adult parasites and poorly on larval stage and immature juvenile worms; this can partially explain the re-infection in endemic areas where patients are likely to host parasites at different developmental stages concurrently. Moreover, the risk of development of drug resistance because of the widespread use of a single drug in a large population is nowadays a serious threat. Hence, research aimed at identifying novel drugs to be used alone or in combination with PZQ is needed. Schistosomes display morphologically distinct stages during their life cycle and epigenetic mechanisms are known to play important roles in parasite growth, survival, and development. Histone deacetylase (HDAC) enzymes, particularly HDAC8, are considered valuable for therapeutic intervention for the treatment of schistosomiasis. Herein, we report the phenotypic screening on both larvae and adult *Schistosoma mansoni* stages of structurally different HDAC inhibitors selected from the in-house Siena library. All molecules have previously shown inhibition profiles on human HDAC6 and/or HDAC8 enzymes. Among them we identified a quinolone-based HDAC inhibitor, NF2839, that impacts larval and adult parasites as well as egg viability and maturation *in vitro*. Importantly, this quinolone-based compound also increases histone and tubulin acetylation in *S. mansoni* parasites, thus representing a leading candidate for the development of new generation anti-Schistosoma chemotherapeutics.

## Author summary

Neglected tropical diseases, such as schistosomiasis, which is caused by parasitic worms, represent serious public health problems in many countries around the world.

**Funding:** This work was supported by the Ministero dell'Università e della Ricerca (MUR), PRIN Project No. 20154JRJPP to G.C. and G.R. and by the CNR (National Research Council)-CNCCS (Collezione Nazionale di Composti Chimici e Centro di Screening) "Rare, Neglected and Poverty Related Diseases Schistodiscovery Project" (DSB.AD011.001.003 to G. R.) and by the Lazio Innova POR FESR Lazio 2014 to 2020 HDACiPLAT Project (grant number A0375-2020-36575 to G. R.) The funders had no role in study design, data collection and analysis, decision to publish, or preparation of the manuscript. RG received salary for postdoctoral fellowship from Lazio Innova grant.

**Competing interests:** The authors have declared that no competing interests exist.

*Schistosoma mansoni* is the major parasitic platyhelminth species causing intestinal schistosomiasis. Currently the control of schistosomiasis is heavily reliant on the drug praziquantel (PZQ) that it is unfortunately not highly effective against larval and immature worm stages. The use of PZQ in mass treatment programs means that the development of resistance is likely, therefore identifying novel drug targets and their associated compounds is critical. Drugs that inhibit enzymes that modify the chromatin structure have been developed for several diseases. We and others have shown that *S. mansoni* epigenetic enzymes are also potential therapeutic targets. Here, we evaluate and characterize several histone deacetylases (HDAC) inhibitors on the larval and adult stage of *Schistosoma mansoni* and report the anti-parasitic profile of these compounds. Some of the HDAC inhibitors tested showed potent effects, particularly NF2839, that impacts larval and adult parasites as well as egg viability and maturation. The findings of this study provide a starting point for the development of new HDAC-based schistosomicidal compounds.

## Introduction

Parasitic diseases remain a major cause of morbidity and mortality globally, mostly affecting people living in the poorest regions of the world. Changes in the environment and population dynamics pose new and global challenges also for schistosomiasis, a neglected tropical disease caused by Platyhelminthes of the genus *Schistosoma*. The three most relevant species for human infections are *Schistosoma mansoni*, *Schistosoma hematobium*, and *Schistosoma japonicum* [1]. Currently, there is no vaccine available to prevent human schistosomiasis, its development being proven to be extremely challenging, mainly for the complexity of the schistosome life cycle and for the parasite's ability to evade the host immune system [2]. Praziquantel (PZQ) is the only approved drug to treat schistosomiasis. PZQ is very effective against adult worms; it is safe and well tolerated, with no significant adverse effects; unfortunately, it is poorly effective on immature stages, such as larval and juvenile worms [3]. PZQ is largely used in mass drug administration programs for the control of schistosomiasis in all endemic settings and development of drug-resistance strains represents a constant threat, also for the appearance of parasite hybrids. Climate changes can further enhance the potential for human and animal populations to encounter new infectious agents, and co-infection by multiple pathogen species within the same host could increase. In the case of helminth parasites, this can occur also through heterospecific (between-species) mate pairings, which can lead to the formation of hybrid offspring. Hybridizations, as well as subsequent introgressions increase genetic diversity and can have important implications in terms of disease control and outcome [4]. Schistosomes are known to hybridize, and natural hybrids are more and more frequently found [5–7]. For all above reasons, new drugs able to target multiple stages of the parasite's life cycle and multiple species are urgently needed. The use of epigenetic drugs including histone deacetylases (HDAC) inhibitors were suggested as a potential novel interesting strategy for antiparasitic therapy including schistosomiasis [8].

  HDACs are members of a family of proteins highly conserved across all eukaryotes. Their main action is to detach acetyl groups from lysine residues at the N-terminal tail of DNA-binding of histones and non-histone proteins [9]. Both Class I (HDACs 1, 3, and 8) and Class II orthologues (HDACs 4, 5, 6/10, 9) have been identified in *S. mansoni* [10–12]. SmHDAC8 is a class I zinc dependent HDAC, which was demonstrated to be abundantly expressed in all *S. mansoni* life cycle stages [10]. Noteworthy, the human homologue hHDAC8 is poorly expressed in human healthy cells and SmHDAC8 is less conserved compared to its human

orthologue than other class I schistosome HDACs 1 and 3 [13]. Therefore, SmHDAC8 has been proposed as a promising target for drug discovery [10]. In recent years, both our group and other research teams have identified HDAC inhibitors that demonstrated effectiveness against *S. mansoni* and/or impacted the enzymatic activity of *S. mansoni* HDAC8 (*Sm*HDAC8) [13–22]. Most discovered inhibitors active on SmHDAC8 present a zinc-binding group, such as a hydroxamate or a sulfur moiety [17]. The general pharmacophoric model of HDAC inhibitors have three key components namely: the surface recognition group (cap), the zinc binding group (ZBG) and a linker bridging the two portions. To achieve better SmHDAC8 activity and selectivity, different structural modifications of the cap and linker have been investigated. Examples of hydroxamic acid-based SmHDAC8 inhibitors reported in literature are shown in Fig 1 [16–18,21–22]

Here to search for novel valuable molecules, we tested structurally different HDAC inhibitors, chosen from the Siena in-house library [23–26]. Fourteen HDAC inhibitors, already recognized to be active on mammalian cells and with a known profile of inhibitory activity on recombinant human HDAC1, HDAC6 and HDAC8, were selected to be tested in phenotypic screenings against *S. mansoni* parasites and SmHDAC8 recombinant protein activity. Selection of novel hits will facilitate further development of novel schistosomicidal molecules.

## Methods

### Ethics statement

Animal work was approved by the National Research Council, Institute of Biochemistry and Cell Biology Animal Welfare Committee (OPBA) and by the competent authorities of the Italian Ministry of Health, DGSAF, Rome (authorization nos. 25/2014-PR and 336/2018-PR). All experiments were conducted in respect to the 3R rules according to the ethical and safety rules and guidelines for the use of animals in biomedical research provided by the relevant Italian law and European Union Directive (Italian Legislative Decree 26/2014 and 2010/63/EU) and the International Guiding Principles for Biomedical Research Involving Animals (Council for the International Organizations of Medical Sciences, Geneva, Switzerland).

### Chemistry

HDAC inhibitors used in this study were prepared as previously reported [23–26].

### Reagents

Chemicals and molecular biology reagents if not otherwise stated were purchased from Merck Life Science Srl (Milan, Italy) or Thermo Fisher Scientific, Italy; Talon superflow histidine-tagged protein purification resin was from GE Healthcare Life Sciences; CellTiter-Glo reagent from Promega; the primary monoclonal anti-α-tubulin antibody (DM1A) and anti-acetylated tubulin (6-11B-1) from Sigma–Aldrich; the anti-acetylated-lysine (Ac-K2-100) from Cell Signaling Technology; the goat anti-mouse and anti-rabbit IgG (H+L) horseradish peroxidase secondary antibodies from Bio-Rad Laboratories. Fluor De Lys HDAC8 Fluorimetric Drug Discovery Kit (BML-AK518) was from Enzo Life Science. Tissue culture media reagents were from Gibco (ThermoFisher Scientific) or EuroClone SpA, Italy.

### Mammalian cell lines and viability assays

BJ cells human fibroblast established from normal foreskin of a neonatal male from the American Type Culture Collection (ATCC; Manassas, VA, USA) (ATCC_CRL-2522) were culture in Minimal Essential Medium (ThermoFisher Scientific, Gibco-11090-081) supplemented with

**Fig 1. Examples of hydroxamic acid based SmHDAC8 inhibitors reported in literature.** J1038, TH65 [16,17]; Compound 9 [22]; NF2624, NF2886 [18,21] are shown.

10% heat-inactivated fetal bovine serum (FBS, EuroClone SpA, Milan, Italy), L-glutammine 2 mM, Sodium Pyruvate 1 mM (GIBCO 11360–039), 0,1 mM non-essential amino acids (NEAA 100x GIBCO- 11140–035) and 100 U/ml of penicillin/streptomycin. The NCTC clone 929 (L-929) murine fibroblast isolated from subcutaneous connective tissue (ATCC-CCL-1) were cultured in Dulbecco's Modified Eagle Medium (DMEM) supplemented with 10 mM Hepes (pH 6.98–7.30), 1 mM l-glutamine, 100 U/ml of penicillin/streptomycin (BioWhittaker, Lonza, EuroClone SpA), and heat-inactivated 10% FBS. All cells were cultured at 37˚C in a 5% $CO_2$ humidified incubator.

The MTT method [27] was employed as previously described to determine the viability of fibroblast cell lines [28]. Briefly, cells seeded into 96-well plates at a density of 20,000 cells/well after overnight incubation, were exposed to selected compounds at the indicated concentrations for 24–72 h. Next they were gently washed, incubated with MTT (bromuro di 3-(4,5-dimetiltiazol-2-il)-2,5-diphenyltetrazolium) for 4 h, and processed for color detection with DMSO. The resulting purple solution was spectrophotometrically measured at 570 nm using a Varioskan Lux instrument and Skanit software (Thermo Fisher Scientific).

The optical density values for both assays were expressed as a percentage of cell survival and normalized with the value of cells treated with vehicle (DMSO), and the data were analyzed using GraphPad Prism v9.5.1 software (San Diego, CA, USA).

### Life cycle *S. mansoni* maintenance, viability assays and egg counts

A Puerto Rican strain of *S. mansoni* was maintained by cycling within albino *Biomphalaria glabrata*, as the intermediate host, and ICR (CD-1) outbred female mice as the definitive host, as previously described [29]. BioWhittaker DMEM lacking phenol red and containing 4500 mg/l glucose, 1 mM Hepes pH 6.98–7.30, 2 mM L-glutamine, 1x antibiotic-antimycotic reagent (Life Technologies) and 10% heat inactivated FBS was used as tissue culture medium for *Schistosoma mansoni* newly transformed schistosoma (NTS). The NTS obtained by

mechanical transformation of cercariae and Percoll gradient was used for the ATP-based luminescence viability assays with the CellTiter-GLO as previously described [29–30]. Briefly, the NTS (150/well) were incubated in DMEM medium (w/o phenol red) complete medium with compounds dissolved in DMSO for 72 h, 37˚C in 5% $CO_2$, in 96-well black plates. Then, 50 μl/well of CellTiter-GLO reagent was added and the relative luminescence unit signal was recorded after 30 min with Varioskan Lux. Data analysis was achieved using GraphPad Prism v9.5.1 software (GraphPad Software Inc) and $LD_{50}$ was obtained after nonlinear regression curve fitting, according to log (inhibitor) *versus* normalized response with a variable slope curve model.

For studies on adult worm pairs, mice infected 7–8 weeks previously with double sex cercariae were euthanized with intraperitoneal injections of Tiletamine/Zolazepam (800 mg/kg) + Xylazine (100 mg/kg) and adult parasites were harvested by reversed perfusion of the hepatic portal system and mesenteric veins. Adult worms were cultured in DMEM containing phenol red and with 4500 mg/l glucose, 2 mM L-glutamine, 100 U/ml penicillin, 100 μg/ml streptomycin, 1x antibiotic-antimycotic reagent, and 10% heat inactivated FBS. Viability assays on adult worm pairs were based on a phenotyping scoring assessment as previously described [29,31]. Briefly, five adult pairs were incubated with selected compounds in 3 ml of DMEM complete medium. For each compound, three experiments were performed, and compounds were given to parasites only once without medium addition and/or replacement. DMSO (vehicle)-treated worms were used as control samples. Viability was monitored under a Leica Model MZ12 stereomicroscope for 7 days and viability scores (0–3) were assigned. In particular, the following scoring criteria were adopted: 3, denoting plate-attached, good movements, clear aspect; 2, representing slower or diminished movements, darkening, minor tegumental damage; 1, denoting movements heavily lowered, darkening, tegument heavily damaged; and 0, meaning dead, lack of any movement. For each sample, the total score was determined by the ratio of the sum of worm scores for the total number of worms.

The eggs produced by worm pairs *in vitro* were counted at day 3 upon treatment using an inverted Leica DM IL microscope (Leica Microsystems, Wetzlar, Germany). Images were captured with a BX41 Olympus (Olympus, Tokyo, Japan) microscope and a brightfield objective 10× served by an Olympus DP23 microscope digital camera and visualized using "CellSens Entry" software (version 3.1.1). Egg maturation morphological score was assigned based on the Vogel and Prata's staging system of egg maturation previously reported [32–33].

## Confocal microscopy analysis

Carmine Red staining was performed as previously described [30]. Images were acquired on a confocal laser scanning TCS SP5 microscope (Leica Microsystems, Wetzlar, Germany), using a 40x (NA = 1.25) oil-immersion lens with optical pinhole at 1AU; Argon laser at 488 nm was used as excitation source and fluorescence was recovered in the range of 500–700 nm. Images were collected as a single stack.

## Western blot analysis

Ten adult pairs were treated with selected HDAC inhibitors for 48 h or 72 h at partially lethal and sublethal concentrations. Compound concentrations were selected based on the viability curve, Trichostatin A (TSA), a pan-HDAC inhibitor, and NF2503 a SmHDAC8 inhibitor chemically characterized by the presence of a hydroxamate-based MBG (metal binding group) coupled to a tricyclic thieno[3,2-b]indole core as capping group, previously described [18] were used as positive controls in the experiments; DMSO (vehicle) was the negative control. Parasites were processed as previously described [18] and samples were analyzed by 15% SDS-PAGE, and immunoblotting was performed using α-tubulin DM1A (1:5000), α-

acetylated-lysine (1:4000) and α-acetylated tubulin (1:3000) antibodies. A Chemidoc XRS Bio-Rad with a chemiluminescent camera and Bio-Rad ImageLab 4.0 software were used for the acquisition and analysis of images. Each Western blot analysis was calibrated by Ponceau staining.

### *E. coli* protein purification and SmHDAC8 activity and inhibition assays

SmHDAC8 protein was produced and purified by a method previously described by us [18]. Inhibition assays of SmHDAC8 were performed as already described in previous publications [18,21]. Briefly, the commercial Fluor de Lys kit (BML-KI178) of Enzo Life Science was used for testing the residual activity of the recombinant enzyme upon preincubation with each of the selected compounds. First the enzyme was preincubated with the selected compounds 15 min before substrate addition to the mixture. The Fluor de Lys substrate was used at 50 μM. Then, compounds, substrate, and enzyme were incubated together, and the reaction was allowed to proceed for 1 h at 30˚C; subsequently, 2 μM TSA within 50 μL of 1× Developer II was added, and the mixture was further incubated for 1 h at 30˚C to quench the reaction. Fluorescence was measured in a plate reader (Varioskan Lux, Thermo Fisher Scientific) with excitation at λ = 370 nm and emission at λ = 450 nm.

### Statistical analysis

All statistical tests were performed using GraphPad Prism version 9.5.1 software (San Diego, CA, USA). The data are shown as mean ± standard deviation (SD) or Standard error of the mean (SEM) as indicated.

## Results and Discussion

Numerous HDAC inhibitors with a clear inhibition profile on human HDAC enzymes showed no effect on the parasite, which may be due to their selectivity for human targets or more likely to poor bioavailability or stability in the parasites; therefore, phenotypic assays on *S. mansoni* are important to identify valuable compounds. Here, we evaluated the effects on NTS (*S. mansoni* larval stage) of a set of inhibitors proven to be active on human class I (hHDAC1 and hHDAC8) and/or class II (hHDAC6) enzymes (Fig 2) and belonging to the in-house Siena library of HDAC inhibitors [23–26].

Compounds **1–3** were chosen among the most selective hHDAC6 inhibitors with an indoline-based cap group and different aromatic and aliphatic moieties at the C2 and C3 positions respectively. These compounds were shown having low inhibitory potency over hHDAC1 and hHDAC8 [23]. Compounds **4–6** were selected among the spiroindoline-based HDAC6 isoform-selective inhibitors [24]. Compounds **7–9** are potent and selective quinolone-based hHDAC6 inhibitors [25]. Compounds **10–14** are dual hHDAC6/hHDAC8 inhibitors with a diphenyl-azetidin-2-one scaffold [26].

### Effects of HDAC inhibitors 1–14 on *S. mansoni* larval stage

The HDAC inhibitors were screened against NTS treated at 50 μM concentration for 72 h to determine their ability to impair viability by using the ATP-based luminescence assay previously developed and validated in our laboratory [29,30]. Two out of 14 compounds, both presenting a quinolone-based cap group, were shown to be effective *in vitro*, decreasing the NTS viability by at least 50%. Next, to calculate the $LD_{50}$ value, dose–response curves were designed by treating parasites with compounds **7** (NF2836) and **9** (NF2839) at concentrations ranging from 0.7 μM up to 100 μM. Compound **8** (NF2838) with a similar structure was assayed as

**Indoline-scaffold (ref 23)**

NF2376 (**1**)

NF2451 **(2)**

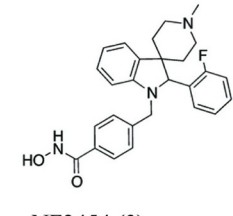

NF2454 **(3)**

**Spiroindoline scaffold (ref. 24)**

NF2657(**4**)

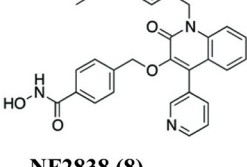

NF2866 **(5)**

NF2870 **(6)**

**Quinolone scaffold (ref. 25)**

**NF2836 (7)**

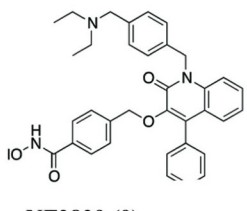

**NF2838 (8)**

**NF2839 (9)**

**Azetidin-2-one scaffold (ref. 26)**

NF2817 (**10**)

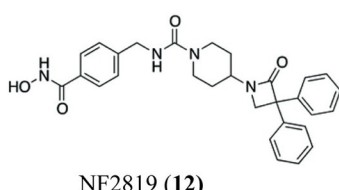

NF2818 **(11)**

NF2819 **(12)**

NF2853 **(13)**

NF2921 **(14)**

**Fig 2. Structure, type of cap, and original reference of HDAC inhibitors tested against *S. mansoni*.**

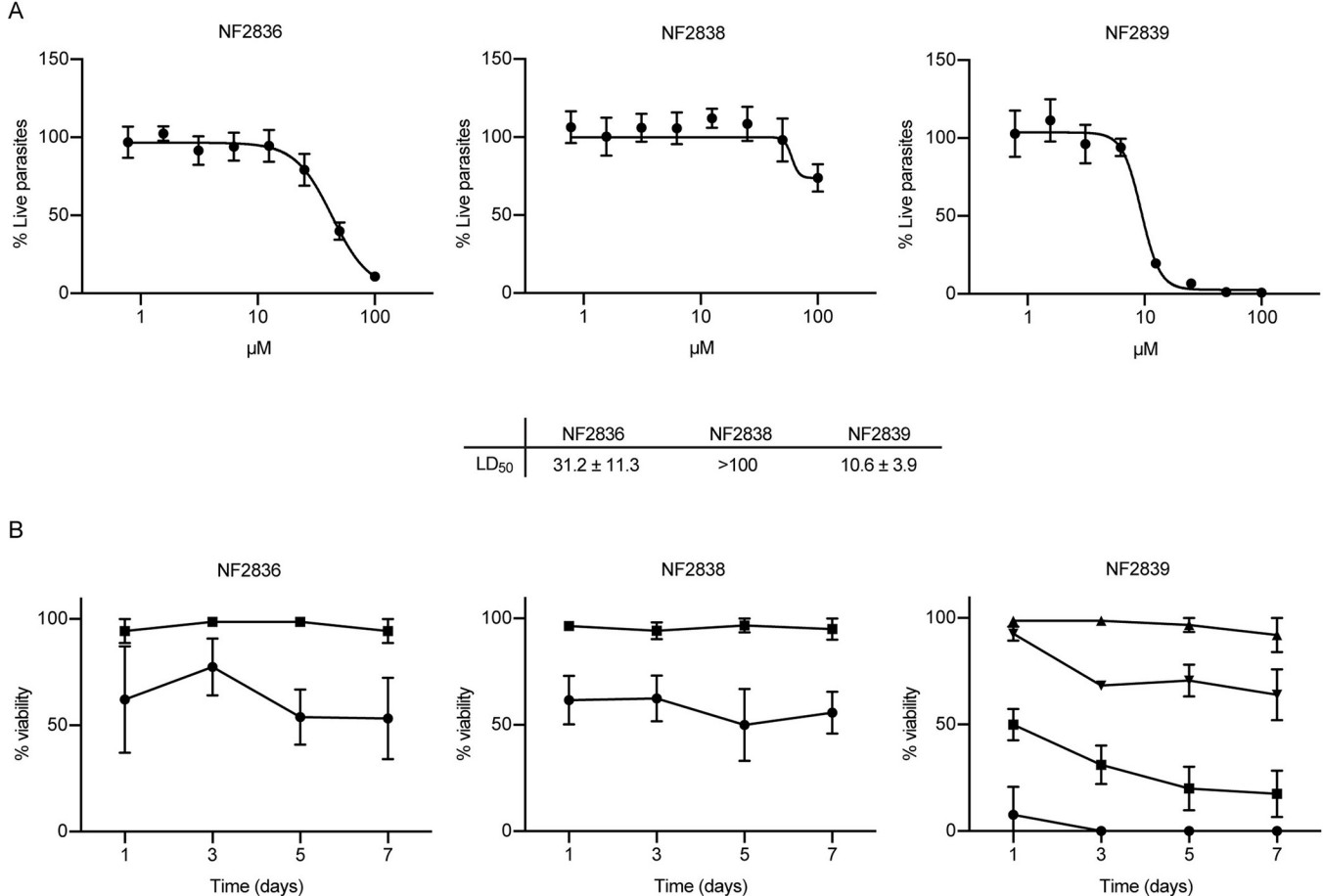

**Fig 3. Schistosome viability assays.** Panel A: Representative dose-response curves of the inhibitors on NTS. The y-axis indicates the percentage of viability normalized against DMSO (100%) and gambogic acid 10 μM (0%). The calculated $LD_{50}$ (μM) average +/- SD are shown. Panel B: dose-response curve of the inhibitors on adult schistosome worm pairs. DMSO (vehicle) was used as negative control (100% viability), the indicated compounds were assayed at 5 (full triangle) 10 (inverted full triangle), 20 (full square) and 50 μM (full circle). Data are expressed as a % severity score (viability) relative to DMSO. Each point represents the average ± standard deviation of three independent experiments.

well at the same concentrations. The $LD_{50}$ values of the three compounds are reported in Fig 3A, along with dose–response curves. The most active compound is NF2839 with a $LD_{50}$ of 10.6 +/- 3.9 μM while the NF2836 shows a lower potency, and the NF2838, with a similar structure, is completely inactive.

## Effects of the quinolone-based compounds on the viability of adult worm pairs, egg production, viability, and maturation

The initial screening of compounds on NTS aimed at identifying promising hit compounds able to impair the larvae viability with a good potency. Since different stages of the schistosome life cycle have different susceptibility to drug actions, the activity of NF2836, NF2838 and NF2839 was investigated also against adult *S. mansoni* worm pairs. Parasites were treated initially at 20 μM and 50 μM; their impact on viability was monitored for 7 days upon treatment. For the most active one, NF2839, parasite's viability was also assessed at lower concentrations 5 and 10 μM. Viability of each schistosome was scored from 3 (no effect) to 0 (severe effects) by assigning daily worm phenotype scores (plate attachment, movement, color, gut peristalsis,

tegument damage) as previously described [29,31] (Fig 3B). Only one of the three compounds markedly decreases the viability of treated worms. NF2839 killed all parasites at 50 μM in 72 h and caused 70% death of *S. mansoni* adult worm pairs in 72 h also at 20 μM. The other two quinolone compounds caused only a 50% reduction in viability at 50 μM; we cannot exclude that their lower efficacy is due to poor bioavailability or stability in our experimental setting or to targeting of HDAC not impacting viability.

The most active compound, NF2839, and one of the less active quinolone compounds, NF2836, in killing adult worm pairs, were then evaluated for their impact on egg production and maturation *in vitro* (Fig 4). First, the highest sublethal concentrations that ensured 50–100% viability and pairing were identified for each compound; hence, egg counts were performed at 72 h upon treatment and normalized to the number of couples. Worm pairs were treated with NF2839 at 5, and 10 μM concentrations; or NF2836 at 20 and 50 μM concentrations. While we did not observe a decrease in the number of eggs laid by worm pairs treated

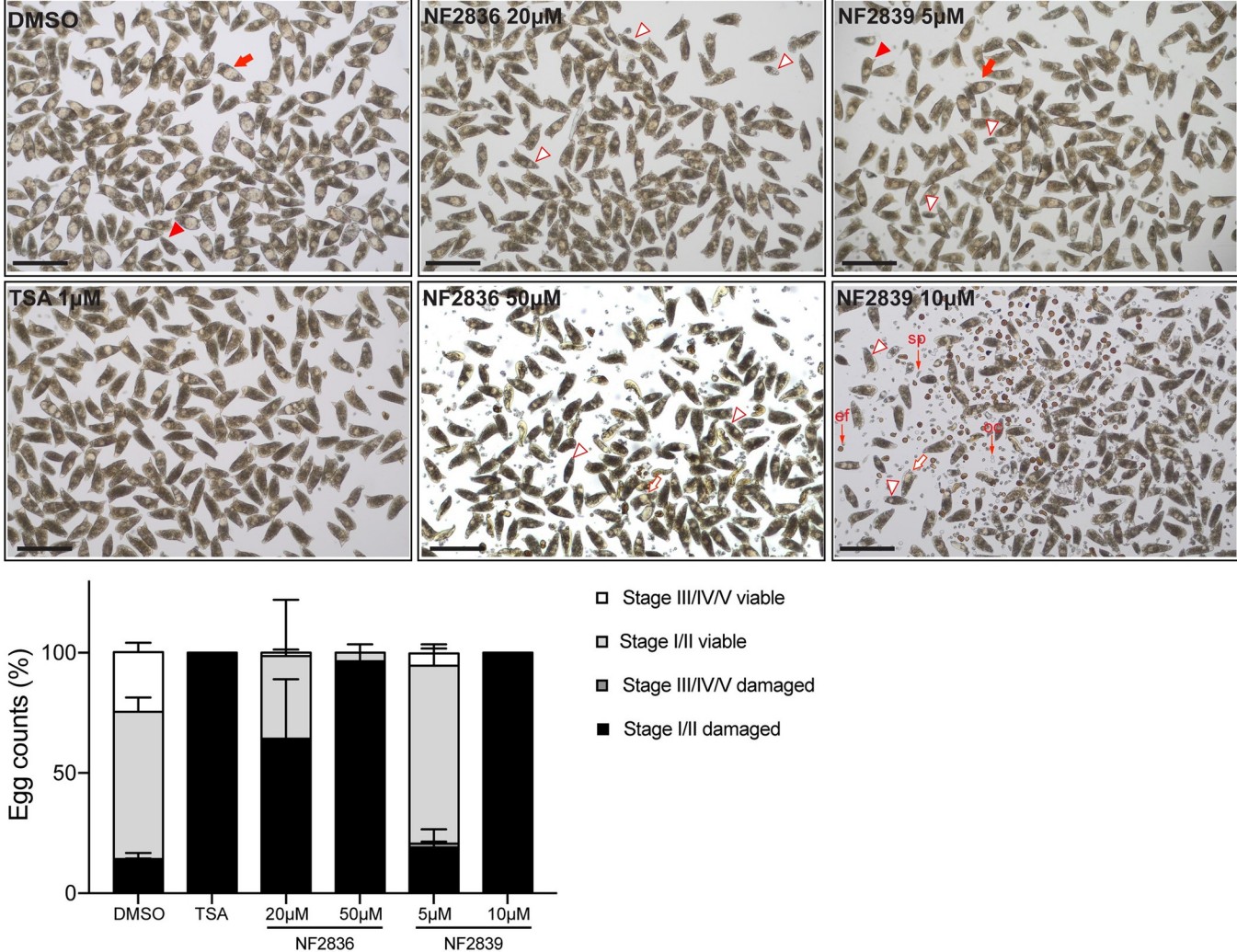

**Fig 4. Quinolone-based HDAC inhibitors impair egg viability and maturation.** Panel A. Representative microscopy images of *S. mansoni* eggs laid by worm pairs exposed to DMSO, NF2836 (20 or 50 μM) or NF2839 (5, 10 μM). Panel B. Histograms represent the percentages of eggs at different maturation stages +/- SEM counted at 72 h after treatment; 5–6 microscopy images of at least 3 independent experiments were analyzed. Filled red triangles indicate viable eggs at stages I–II (immature); empty red triangles indicate damaged/morphological abnormal eggs at stages I–II (immature); filled red arrows indicate viable eggs at stages III–V (intermediate/developed); empty red arrows indicate damaged/morphological abnormal eggs at stages III-V; ef, egg fragments; sp, sperm cells; oc, oocyte. Scale Bars = 200 μm.

with the inhibitors, the eggs showed clear signs of damage and defects in maturation when compared with the vehicle control samples. Moreover vitelline, germ cells and egg fragments were present in the culture medium. Therefore at 72 h upon worm pairs treatment the eggs were counted and classified by microscopic observation according to the Vogel and Prata staging system of egg maturation [32,33] and grouped as immature (stages I and II), intermediate/developed (stages III–V), viable or damaged eggs. The samples treated with vehicle (DMSO) developed as expected. A drastic effect of NF2836 and NF2839 compounds on egg maturation was observed; essentially all eggs were arrested in stage I/II. Moreover, following treatment with the compounds, eggs appeared seriously damaged and not viable (Fig 4).

To characterize the phenotypic alterations induced by compound treatment in more detail, carmine-red staining, followed by confocal laser scanning microscopy (CLSM) analysis was performed (Fig 5). Sublethal doses of compounds were chosen to preserve the overall parasite viability and pairing. For each compound, 3–5 couples in two independent experiments were analyzed and images of ootype, ovary, vitellarium, testis, gut, and parenchyma were recorded. The overall morphology of the ovary in worm pairs treated with NF2836 and NF2839 compounds seemed not to be affected thus retaining the organization with small immature oocytes (io) in the anterior part and large mature oocytes (mo) in the posterior part. However, a decrease of cellularity in the ovary mature compartment was observed in 5/11 samples treated with NF2836 at 50 uM; cell degeneration especially in the mature compartment was detected in the oocytes of all NF2839-treated worms, with an increase in the presence of black spots in the worms treated with 10 μM (Fig 5). The vitellarium retains a preserved structure in all analyzed samples. Regarding the ootype, worm pairs treated with both compounds at the higher concentration showed either an empty ootype or no properly developed eggs and/or egg fragments, while at the lower concentrations, 5/7 or 4/7 ootype contained damaged eggs respectively in NF2836- or NF2839-treated worms (Fig 5). In addition, treatment with NF2836 and NF2839 compounds at the higher concentration led in all worms, in particular males, to gut dilation with a dramatic decrease in the number of surface amplification and lumen epithelial thinning. Accumulation of carmine-red positive particle aggregates was also present in the lumen of the NF2839-treated male worms (Fig 5).

## Effects of the quinolone-based compounds on adult worm pairs histone and tubulin acetylation

To investigate the HDAC inhibitory activity in adult parasites, cytosolic and nuclear-enriched fractions of worm pairs treated for 48 h and 72 h with partially lethal and sub-lethal doses of NF2836, NF2838, and NF2839 compounds were analyzed by western blotting (Fig 6). The samples treated with the three compounds showed increased acetylation of both histone proteins and tubulin when compared with vehicle (DMSO). TSA, a known pan- class I and class II HDAC -inhibitor, and NF2503, a HDAC inhibitor previously shown to impact histone acetylation in adult worm pairs [18] were used as positive controls. NF2839 and NF2838 treatment more strongly impact histone acetylation, while NF2836, which only slightly impacts histone acetylation, induces the highest increase in tubulin acetylation (Fig 6).

## Activity of hit compounds on recombinant *Sm*HDAC8

All three quinolone-based compounds inhibited the activity of the recombinant form of SmHDAC8 at the low micromolar range (Table 1). Therefore, these HDAC inhibitors have a deacetylase inhibitory activity in parasites and are also targeting SmHDAC8. The increase in tubulin acetylation in cell lysates of HDAC inhibitors-treated worm pairs (Fig 6) and their nanomolar inhibitory activity on hHDAC6 (Table 1) point to the possibility that others Class I

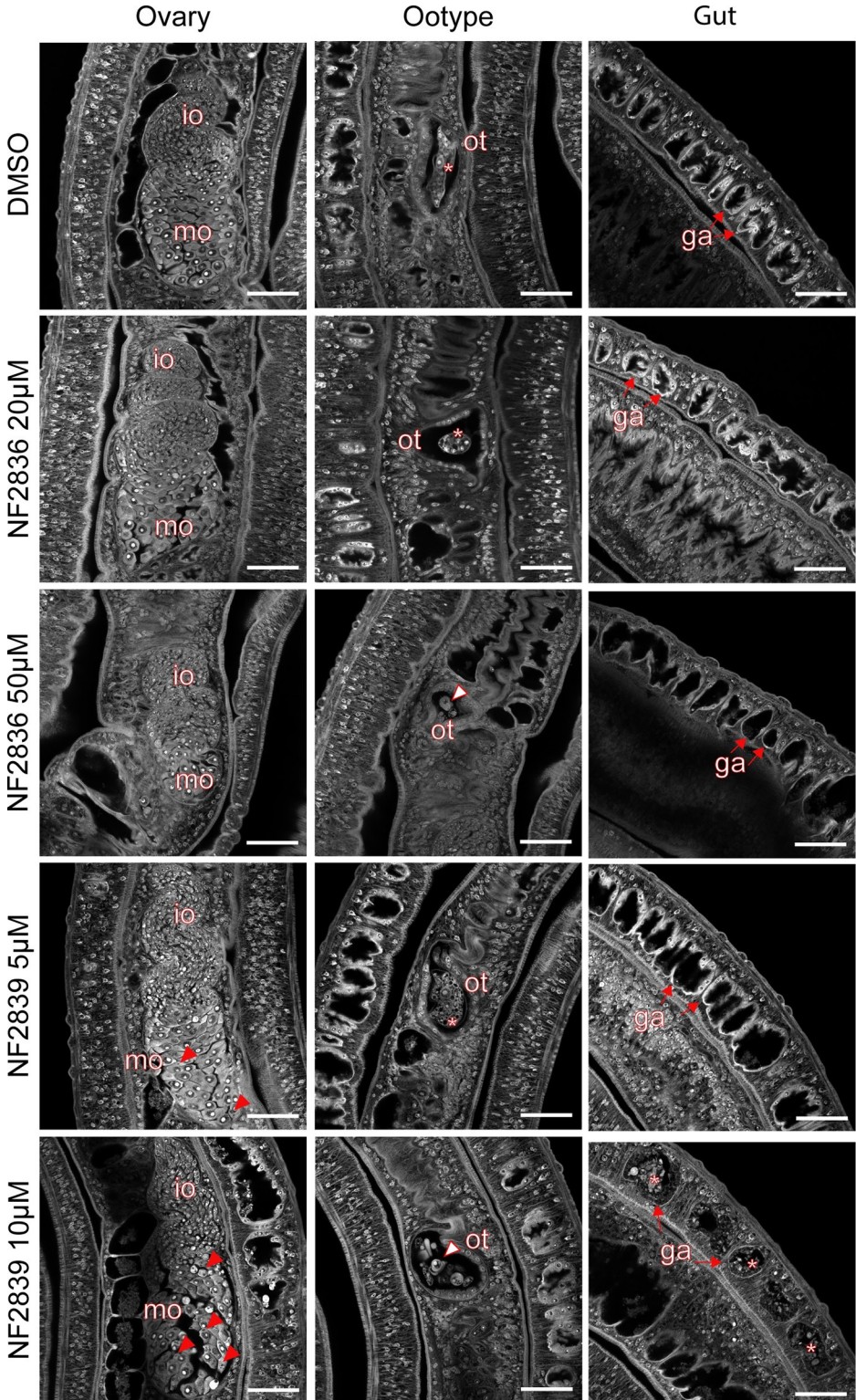

**Fig 5. Confocal microscopy of carmine-red stained adult worm pairs treated with quinolone-based HDAC inhibitors.** The images are representative of 3–4 worm pairs treated with DMSO, NF2836 (20 and 50 μM) or NF2839 (5 and 10 μM) at 72 h in two independent experiments. The ovary, ootype and gut are shown. Immature oocytes (io), mature oocytes (mo), ootype (ot) and gastrodermis (ga) are labeled. In the Figure, red filled triangles indicate degenerated mo; empty red triangles indicate egg fragments in the ootypes; asterisks indicate deformed egg in the ootype and carmine red aggregates in the gut lumen, filled red arrows indicate the ga in the gut. Scale bars: 50 μm.

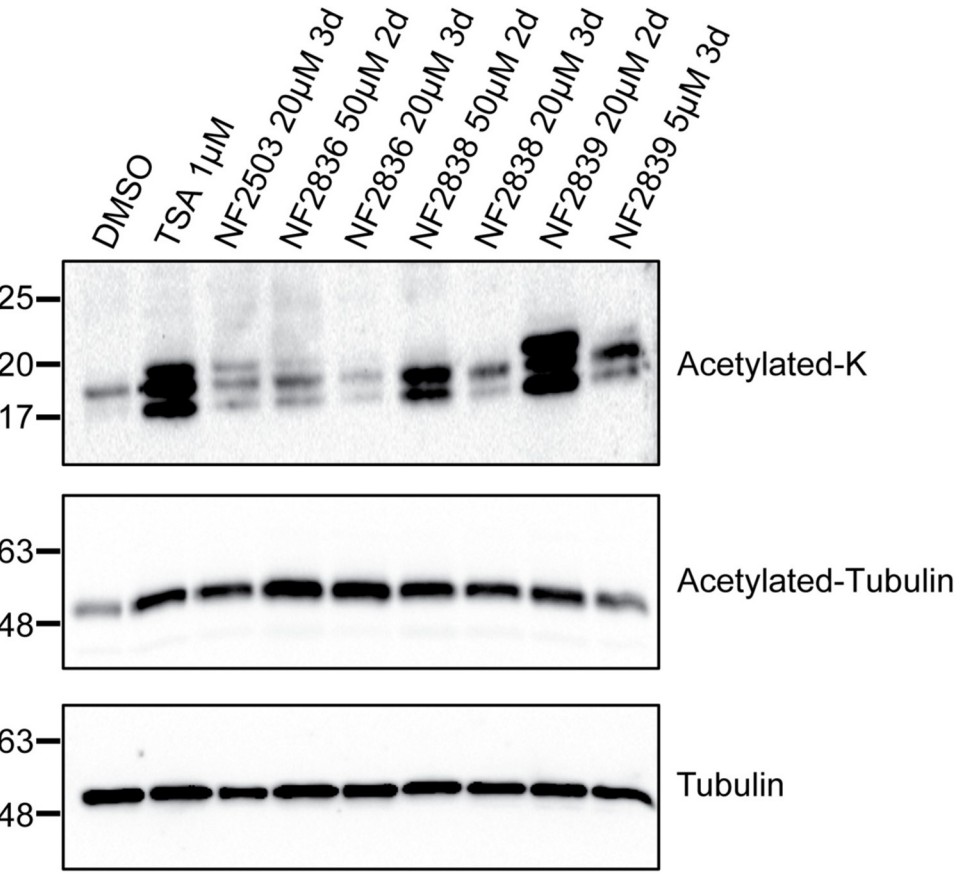

**Fig 6. Effects of selected compounds on histones and tubulin acetylation.** Representative immunoblots of the histone-enriched and cytosolic protein fractions extracted from *S. mansoni* adult worm pairs incubated with anti-acetylated lysine antibody (acetylated-K) or anti-acetylated tubulin. Worm pairs were treated for 48 or 72 h at the concentration indicated for each compound. DMSO (vehicle) and of the HDAC pan- inhibitor, TSA (1 μM), were used respectively as negative and positive controls.

or II HDACs could be targeted. Interestingly non-selective benzanilide HDAC inhibitors with very little, to no-observed potency against both human and *S. mansoni* HDAC8 orthologues and schistosomicidal activity have been recently described [22]. Overall, we cannot exclude that the NF2839 and the other quinolone-based compounds are impacting parasite viability and/or eggs viability and maturation also through engagement of other HDACs. Moreover, off-targets effects cannot be excluded.

**Table 1. Activity of quinolone-based compounds on SmHDAC8 activity.**

| HDAC Inhibitor | hHDAC1* (μM) | hHDAC6* (nM) | hHDAC8* (μM) | SmHDAC8 (μM) |
|---|---|---|---|---|
| NF2836 | 3.0 ± 0.2 | 33.0 ± 2.6 | 1.6 ± 0.3 | 2.3 ± 0.8 |
| NF2838 | 0.3 ± 0.1 | 11.5 ± 1.5 | 0.5 ± 0.2 | 0.4 ± 0.03 |
| NF2839 | 0.3± 0.1 | 6.9 ± 0.9 | 0.6 ± 0.2 | 0.1± 0.3 |

The $IC_{50}$ of the indicated inhibitors on recombinant SmHDAC8 enzymatic activity are shown as an average of three independent experiments; uncertainties are given as standard deviations. The Fluor-de-Lys substrate was used at 50 μM, and the inhibitor concentrations ranged from 25 μM to 0.04 μM. The $IC_{50}$ on recombinant hHDAC1, hHDAC6 and hHDAC8 enzymatic activity previously reported [25] are also listed.

### In silico prediction of selected drug-like properties and cytotoxicity evaluation of the hit compounds

The ADME properties and the heart toxicity potential of the compounds by means of QikProp software (Version 4.3, Schrodinger, LCC, New York, NY 2015) have been previously reported [25] and are summarized in S1 Table. The predicted values can be considered acceptable and were within the recommended range.

The cytotoxicity of the hit compounds was assessed in two independent experiments on murine L929 and human BJ fibroblasts at 24 h by MTT assays. Vehicle (DMSO) and Gambogic acid (GA) were used as negative and positive control respectively. The compounds show a safe profile on both cell lines (S2 Table).

## Conclusions

We discovered a schistosomicidal activity of a quinolone-based HDAC inhibitor, NF2839, on both larval and adult worm pairs also impacting egg viability and maturation *in vitro*. Importantly, this compound increases histone and tubulin acetylation in *S. mansoni* parasites, representing a candidate worthy of further consideration as starting point for the development of new generation anti-*Schistosoma* chemotherapeutics.

## Supporting information

**S1 Data. Excel spreadsheet containing, in separate sheets, the underlying numerical data and statistical analysis for Figs 3A, 3B, and 4.**
(XLSX)

**S1 Table. Predicted drug-like features for NF2836, NF2838 and NF2839 compounds.**
[a]SASA predicted the total solvent accessible surface (range or recommended value for 95% of known drugs 300–1000); [b]QPlogP predicted octanol/water partition coefficient (range or recommended value for 95% of known drugs -2–6.5); [c]QPlogS predicted aqueous solubility in mol/dm$^3$(range or recommended value for 95% of known drugs -6.5–0.5); [d]QPPCaco predicted apparent Caco-2 cell permeability in nm/sec (range or recommended value for 95% of known drugs >500 great); [e]QPPMDCK predicted apparent MDCK cell permeability in nm/sec (range or recommended value for 95% of known drugs >500 great); [f]QPlogHERG predicted IC$_{50}$ values for blockage of HERG K$^+$ channels (range or recommended value for 95% of known drugs below -5); [g]%HOA predicted human oral absorption on 0 to 100% scale (range or recommended value for 95% of known drugs >80% high). Range or recommended values are reported in QikProp user manual. *Cmpd name reported in Relitti et al [25].
(DOCX)

**S2 Table. Cytotoxicity for NF2836, NF2838 and NF2839 compounds on BJ and L929 fibroblast cell lines.** The IC$_{50}$(μM) shown was calculated by dose-response curves of the compounds on murine L929 and human BJ fibroblast cells. The range of concentrations were for NF2836, NF2838, NF2839: 100–0.78 μM; and for GA 10–0.078 μM. The data are representative of 2 independent experiments. Gambogic acid (GA) and vehicle (DMSO) were used as positive and negative controls respectively.
(DOCX)

## Acknowledgments

We wish to thank Dr. Marialaura Giannaccari, IBBC-CNR for her support on SmHDAC8 recombinant protein purification, Stefania Colantoni for mouse husbandry and Pierluigi Palozzo for dishwashing and lab technical support.

## Author Contributions

**Conceptualization:** Giovina Ruberti.

**Data curation:** Roberto Gimmelli, Giuliana Papoff, Fulvio Saccoccia, Cristiana Lalli, Sandra Gemma, Giuseppe Campiani, Giovina Ruberti.

**Formal analysis:** Roberto Gimmelli, Giuliana Papoff, Fulvio Saccoccia, Cristiana Lalli, Sandra Gemma, Giuseppe Campiani, Giovina Ruberti.

**Funding acquisition:** Giuseppe Campiani, Giovina Ruberti.

**Investigation:** Roberto Gimmelli, Giuliana Papoff, Fulvio Saccoccia, Cristiana Lalli, Sandra Gemma.

**Methodology:** Roberto Gimmelli, Giuliana Papoff, Fulvio Saccoccia, Cristiana Lalli.

**Supervision:** Giovina Ruberti.

**Validation:** Roberto Gimmelli, Giuliana Papoff, Fulvio Saccoccia.

**Visualization:** Roberto Gimmelli, Giuliana Papoff, Fulvio Saccoccia, Sandra Gemma.

**Writing – original draft:** Giovina Ruberti.

**Writing – review & editing:** Roberto Gimmelli, Giuliana Papoff, Fulvio Saccoccia, Sandra Gemma, Giuseppe Campiani, Giovina Ruberti.

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
