## [Decision Letter · Decision Letter 0]

6 Nov 2023

Dear Dr Ruberti,

Thank you very much for submitting your manuscript "Effects of structurally distinct Human HDAC6 and HDAC6/HDAC8 inhibitors against S. mansoni larval and adult worm stages" for consideration at PLOS Neglected Tropical Diseases. As with all papers reviewed by the journal, your manuscript was reviewed by members of the editorial board and by several independent reviewers. In light of the reviews (below this email), we would like to invite the resubmission of a significantly-revised version that takes into account the reviewers' comments. 

We cannot make any decision about publication until we have seen the revised manuscript and your response to the reviewers' comments. Your revised manuscript is also likely to be sent to reviewers for further evaluation.

Sincerely,

Aaron R. Jex

Section Editor

Aaron Jex

Section Editor

Reviewer's Responses to Questions

**Key Review Criteria Required for Acceptance?**

**Methods**

-Are the objectives of the study clearly articulated with a clear testable hypothesis stated?

-Is the study design appropriate to address the stated objectives?

-Is the population clearly described and appropriate for the hypothesis being tested?

-Is the sample size sufficient to ensure adequate power to address the hypothesis being tested?

-Were correct statistical analysis used to support conclusions?

-Are there concerns about ethical or regulatory requirements being met?

Reviewer #1: Methods are OK.

**Results**

-Does the analysis presented match the analysis plan?

-Are the results clearly and completely presented?

-Are the figures (Tables, Images) of sufficient quality for clarity?

Reviewer #1: Results are OK except for a misunderstanding/misreporting of significant figures.

**Conclusions**

-Are the conclusions supported by the data presented?

-Are the limitations of analysis clearly described?

-Do the authors discuss how these data can be helpful to advance our understanding of the topic under study?

-Is public health relevance addressed?

Reviewer #1: Please see comments below.

**Editorial and Data Presentation Modifications?**

Reviewer #1: Please see comments below.

**Summary and General Comments**

Reviewer #1: 1. In light of the following sentence in the introduction section:” In recent years, both our group and other research teams have identified HDAC inhibitors that demonstrated effectiveness against S. mansoni and/or impacted the enzymatic activity of S. mansoni HDAC8 (SmHDAC8), (13–20).” the authors should include chemical structures of these ‘identified HDAC inhibitors’ in a new figure. 

2. How do the authors justify four significant figures for their reported LD50 values in Figure 1?

3. The following sentence: “However, we cannot exclude that the NF2839 and the other quinolone-based compounds are impacting parasite viability and/or eggs viability and maturation also through inhibition of other HDACs.” unjustifiably assumes that all of these compounds are inhibiting various aspects of schistosome biology solely through HDAC inhibition. These compounds, very likely, are acting by multiple mechanisms, only one of which may be HDAC inhibition. For example, was there any correlation between HDAC inhibition and schistosome inhibition for these compounds?

4. A consistent number of significant figures should be used for the data in Table 2. 

5. The authors should include cytotoxicity data for all of the compounds reported in this study. Without these data, the reader has no idea if these compounds have any antischistosomal selectivity. 

6. Did the authors generated any in vitro DMPK data for these compounds?

PLOS authors have the option to publish the peer review history of their article (what does this mean?). If published, this will include your full peer review and any attached files.

Reviewer #1: No
---

## [Decision Letter · Decision Letter 1]

24 Jan 2024

Dear Dr Ruberti,

Thank you very much for submitting your manuscript "Effects of structurally distinct Human HDAC6 and HDAC6/HDAC8 inhibitors against S. mansoni larval and adult worm stages" for consideration at PLOS Neglected Tropical Diseases. As with all papers reviewed by the journal, your manuscript was reviewed by members of the editorial board and by several independent reviewers. The reviewers appreciated the attention to an important topic. Based on the reviews, we are likely to accept this manuscript for publication, providing that you modify the manuscript according to the review recommendations. 

Sincerely,

Aaron R. Jex

Section Editor

Aaron Jex

Section Editor

Reviewer's Responses to Questions

**Key Review Criteria Required for Acceptance?**

**Methods**

-Are the objectives of the study clearly articulated with a clear testable hypothesis stated?

-Is the study design appropriate to address the stated objectives?

-Is the population clearly described and appropriate for the hypothesis being tested?

-Is the sample size sufficient to ensure adequate power to address the hypothesis being tested?

-Were correct statistical analysis used to support conclusions?

-Are there concerns about ethical or regulatory requirements being met?

Reviewer #1: (No Response)

**Results**

-Does the analysis presented match the analysis plan?

-Are the results clearly and completely presented?

-Are the figures (Tables, Images) of sufficient quality for clarity?

Reviewer #1: Despite their claims in the reponse to the initial manuscript review, the authors did not correct their inconsistent and faulty use of significant figures.

**Conclusions**

-Are the conclusions supported by the data presented?

-Are the limitations of analysis clearly described?

-Do the authors discuss how these data can be helpful to advance our understanding of the topic under study?

-Is public health relevance addressed?

Reviewer #1: (No Response)

**Editorial and Data Presentation Modifications?**

Reviewer #1: (No Response)

**Summary and General Comments**

Reviewer #1: (No Response)

PLOS authors have the option to publish the peer review history of their article (what does this mean?). If published, this will include your full peer review and any attached files.

Reviewer #1: No

Figure Files:

Data Requirements:

Reproducibility:

References

---

## [Editor Report · Decision Letter 2]

13 Feb 2024

Dear Dr Ruberti,

We are pleased to inform you that your manuscript 'Effects of structurally distinct Human HDAC6 and HDAC6/HDAC8 inhibitors against S. mansoni larval and adult worm stages' has been provisionally accepted for publication in PLOS Neglected Tropical Diseases.

Best regards,

Aaron R. Jex

Section Editor

Aaron Jex

Section Editor

---

## [Editor Report · Acceptance letter]

21 Feb 2024

Dear Dr Ruberti,

We are delighted to inform you that your manuscript, "Effects of structurally distinct Human HDAC6 and HDAC6/HDAC8 inhibitors against S. mansoni larval and adult worm stages," has been formally accepted for publication in PLOS Neglected Tropical Diseases.

Best regards,

Shaden Kamhawi

co-Editor-in-Chief

Paul Brindley

co-Editor-in-Chief
